# Atypical memory B-cells and autoantibodies correlate with anemia during *Plasmodium vivax* complicated infections

Juan Rivera-Correa[1][¤]*, Maria Fernanda Yasnot-Acosta[2], Nubia Catalina Tovar[1,2,3,4], María Camila Velasco-Pareja[2], Alice Easton[1], Ana Rodriguez[1]*

**1** New York University School of Medicine, New York, United States of America, **2** Grupo de Investigaciones Microbiológicas y Biomédicas de Córdoba, Universidad de Córdoba, Colombia, **3** Universidad del Sinú, Montería, Colombia, **4** Universidad de Cartagena, Bolívar, Colombia

☯ These authors contributed equally to this work.
¤ Current address: Autoimmunity and Inflammation Program, Hospital for Special Surgery, New York, United States of America
* jlblue55@gmail.com (JRC); Ana.Rodriguez@nyumc.org (AR)

## Abstract

Malaria caused by *Plasmodium vivax* is a highly prevalent infection world-wide, that was previously considered mild, but complications such as anemia have been highly reported in the past years. In mice models of malaria, anti-phosphatidylserine (anti-PS) autoantibodies, produced by atypical B-cells, bind to uninfected erythrocytes and contribute to anemia. In human patients with *P. falciparum* malaria, the levels of anti-PS, atypical B-cells and anemia are strongly correlated to each other. In this study, we focused on assessing the relationship between autoantibodies, different B-cell populations and hemoglobin levels in two different cohorts of *P. vivax* patients from Colombia, South America. In a first longitudinal cohort, our results show a strong inverse correlation between different IgG autoantibodies tested (anti-PS, anti-DNA and anti-erythrocyte) and atypical memory B-cells (atMBCs) with hemoglobin in both *P. vivax* and *P. falciparum* patients over time. In a second cross-sectional cohort, we observed a stronger relation between hemoglobin levels, atMBCs and autoantibodies in complicated *P. vivax* patients compared to uncomplicated ones. Altogether, these data constitute the first evidence of autoimmunity associating with anemia and complicated *P. vivax* infections, suggesting a role for its etiology through the expansion of autoantibody-secreting atMBCs.

## Author summary

Malaria is one of the top global infections causing high mortality and morbidity every year. *Plasmodium vivax* is the most prevalent malarial infection, particularly in the region of the Americas. Complications associated with *P. vivax*, such as anemia, are a growing reported phenomenon, but the mechanisms leading to them are poorly understood. Here, we report the first evidence of autoantibodies and Atypical Memory B-cells correlating with anemia in two different cohorts of *P. vivax* patients, particularly during complicated

**Data Availability Statement:** All relevant data are within the manuscript and its Supporting Information files.

**Funding:** Funding was obtained from National Institutes of Health Training Grant T32 AI007180 to J.R.C., Ministerio de Ciencia Tecnología e Innovación de Colombia, Convocatoria Doctorados Nacionales 727 (2015) to N.C.T. and Universidad de Cordoba Proyecto FCS 01-17 to M.F.Y.A. The funders had no role in study design, data collection and analysis, decision to publish, or preparation of the manuscript.

**Competing interests:** The authors have declared that no competing interests exist.

infections. These findings point to Atypical Memory B-cells as key pathological players, possibly through the secretion of autoantibodies, and attributes a role for autoimmunity in mediating complications during *P. vivax* infections.

## Introduction

*Plasmodium vivax* is the predominant cause of malaria in many areas of the world, including South and Central America, where it represents 75% of malaria cases [1]. *P. vivax* malaria was traditionally considered a low-risk uncomplicated infection, but in the past years an increasing number of reports have documented severe complications and death caused by this infection [2–4]. Complications of *P. vivax* infections include different manifestations, but severe anemia is among the most frequent, especially in children [5, 6]. Despite its growing prevalence, the mechanisms leading to complications during *P. vivax* infections are poorly understood.

Anemia in malaria is a multifactorial syndrome characterized by decreased erythropoiesis and by the loss of infected and uninfected erythrocytes [7, 8], which results in the loss of about 34 uninfected erythrocytes for each erythrocyte lysed directly due to *P. vivax* infection [9]. The mechanisms underlying the loss of uninfected erythrocytes are not clear yet, but malaria-induced anemia was recently related to autoimmune responses in patients [10]. Malaria, as other highly inflammatory infectious diseases, induces a strong autoimmune response characterized by the generation of anti-self antibodies with different specificities [11–13]. Studies in mice models of malaria showed that antibodies recognizing the lipid phosphatidylserine (PS) exposed on the surface of uninfected erythrocytes promote their clearance contributing to anemia [14].

In malaria patients, the levels of anti-PS antibodies correlate inversely with hemoglobin levels in different cohorts infected with *P. falciparum*, including children with severe infections in Uganda [15], European travelers with post-malarial anemia [14] or first-time malaria infections [16] and uncomplicated *P. vivax* infections in Malaysia [17]. The relationship between anti-PS antibodies and other autoantibodies with anemia has not been explored longitudinally or during complicated *P. vivax* infections. We hypothesized that anti-PS and other autoantibodies would correlate with anemia development during *P. vivax* malaria, particularly in complicated infections.

Previous reports show increased levels of atypical memory B-cells (AtMBCs) in populations chronically exposed to *P. falciparum* [17–21] or *P. vivax* infections [22]. In *P. falciparum* acute infections, a strong correlation was observed between the levels of AtMBCs, the levels of anti-PS antibodies and the levels of plasma hemoglobin [16], suggesting that atMBCs may be the main B-cell type secreting anti-PS antibodies that contribute to human malarial anemia, as was previously observed in mice models of infection [23]. However, the relationship between AtMBCs, autoimmunity and the role they might play during anemia and other complications has not been explored during *P. vivax* infections. We hypothesized that AtMBCs would be highly expanded during complicated *P. vivax* infections and could be a key mediator of anemia though the secretion of autoimmune antibodies.

Here we present the first study of the relations between autoimmune antibodies, hemoglobin levels and AtMBCs in two different cohorts of *P. vivax* malaria patients from Colombia: one longitudinal comparing uncomplicated *P. vivax* and *P. falciparum* patients over the period of one month and one cross-sectional comparing complicated and uncomplicated *P. vivax* malaria.

Our results from the first cohort show that the levels of autoimmune antibodies and AtMBCs are maintained at least during one month after infection and correlate with anemia in both *P. vivax* and *P. falciparum* patients. In the second cohort, we analyzed the relations of different clinical and immune parameters of patients with uncomplicated or complicated *P. vivax* infections. A correlation analysis revealed a relation between autoimmune antibodies and hemoglobin levels in patients with complicated *P. vivax* infections, which were also related to levels of AtMBCs.

## Methods

### Ethics statement

Both studies included in this manuscript were approved by the Committee on Human Ethics of the Health Sciences Department of the University of Cordoba, Monteria, Colombia in Acta #004 on May 6, 2016. Written informed consent was received from participants prior to inclusion in the study.

### Study design and sample collection

Two different studies on patients with malaria are included in this work (Table 1). The patients from both cohorts were recruited at the Tierralta municipality (8˚10′22″N 76˚03′34″O) in Córdoba, Colombia. This municipality expands for a total of 5.025 Km$^2$, has an average temperature of 27.3˚C and an altitude of 51 m (S1 Fig). The malaria incidence in Tierralta is characterized for having stable transmission and high risk across the year, with an annual parasitological index above 10 cases per 1,000 habitants. In 2019, 9,111 malaria cases were reported, with 0.3% of them categorized as complicated malaria. *P. vivax* is the dominant species being reported, with a ratio of 4:3 *P. vivax* to *P. falciparum* [24]. Patients for the first cohort to follow autoimmune responses over time were recruited at Hospital San José of Tierralta, Córdoba, Colombia, during six months in 2017 (Table 2). Uninfected controls were recruited in the urban, malaria non-endemic area of Monteria [24]. Patients for the second cohort to compare uncomplicated and complicated *P. vivax* infections were recruited at Hospital San Jerónimo of Monteria and Hospital San José of Tierralta, Córdoba, Colombia, between October 2017 and March 2019 (Table 3). For both studies inclusion criteria were diagnosis of *P. vivax* by blood smear, confirmed by nested PCR [25]. The WHO criteria for diagnosis of anemia [26] and for severe *P. vivax* were followed [6]. The most frequent complications in this group were thrombocytopenia (platelets $\leq$ 50,000/μL) in 64% (32/50) of patients; high alanine aminotransferase levels ($>$ 40 U/L) in 48% (24/50) of patients and hypoglycemia (glucose $\leq$ 60 mg/dL) in 42% (21/50) of patients. Only one patient presented severe anemia (hemoglobin $\leq$ 8 g/dL) while most patients suffered from moderate to mild anemia (Table 4). In both cohorts, patients were treated according to the National Health Institute of Colombia guidelines: Chloroquine

**Table 1.  Description of cohorts 1 and 2.**

|  | Cohort 1 | Cohort 2 |
|---|---|---|
| **Study type** | Longitudinal | Cross-sectional |
| ***Plasmodium* species** | *P. vivax* (n = 11) *P. falciparum* (n = 9) | *P. vivax* |
| **Community controls (CC)** | 8 | 50 |
| **Uncomplicated malaria (UM)** | 20 | 56 |
| **Complicated malaria (CM)** | - | 50 |
| **Follow up days post treatment** | 0, 7, 14, 21, 28 | - |

**Table 2. Clinical information from community controls, *P. vivax* and *P. falciparum*-infected patients from cohort 1.**

| | CC (n = 8) | *P. vivax* (n = 11) | *P. falciparum* (n = 9) | *P value Pv vs Pf | *P value CC vs Pv | *P value CC vs Pf |
|---|---|---|---|---|---|---|
| Age (years) | 25.5 (21.5, 28.7) | 21 (13.0, 23,0) | 13 (10.0, 21.5) | - - - | - - - | - - - |
| Female sex (%) | 6 (75) | 4 (36.3) | 5 (55.5) | - - - | - - - | - - - |
| Hemoglobin (g/dl)* | 13.4(13.53,14) | 11.8 (11.0, 12.1) | 11.2 (10.5, 12.0) | 0.5389 | <0.0001 | 0.0013 |
| Parasite density (p/ μl) | 0 (0, 0) | 5,903 (4,913; 8,752) | 7,733 (3,988; 8,865) | 0.8633 | - - - | - - - |

*Hemoglobin levels at nadir in *P. falciparum* and *P. vivax* patients.

Data presented as median (IQR) unless otherwise indicated

Abbreviations: community controls (CC), *P. vivax (Pv)*, *P. falciparum (Pf)*,

**Table 3. Clinical information from community controls, *P. vivax* uncomplicated and complicated patients from cohort 2.**

| | CC (n = 50) | UM *P. vivax* (n = 56) | CM *P. vivax* (n = 50) | *P value UM vs CM |
|---|---|---|---|---|
| Age (years) | 20.5 (13.0, 33.2) | 17.5 (12.7, 33.7) | 16.0 (11.0, 26.0) | - - - |
| Female sex (%) | 24 (48.0) | 22 (39.3) | 24 (48.0) | - - - |
| Hemoglobin (g/dl) | 12.6 (11.5, 13.7) | 11.5 (10.3, 12.7) | 11.2 (9.9, 12.5) | 0.5643 |
| Parasite density (p/μl) | 0 (0, 0) | 2,387 (1,800, 4050) | 2,500 (1,460,5,185) | 0.1421 |

Data presented as median (IQR) unless otherwise indicated

Abbreviations: community controls (CC); uncomplicated *P. vivax* malaria (UM); complicated *P. vivax* malaria (CM).

(10 mg/kg, followed by 7.5 mg/kg at 24 and 48 h) and Primaquine (0.25 mg/kg for 14 days) [27]. For all groups, children younger than 2 years old, pregnant women, and patients with other non-malarial infections (and *P. falciparum* for the cohort 2), were excluded. The following infections were excluded: Dengue, brucellosis, leptospirosis, salmonellosis, rickettsial disease and mixed *Plasmodium* infection. Control subjects for the second cohort were recruited in the municipality of Tierralta among afebrile people with no malaria episodes in the past 6 months. All were confirmed to be PCR negative for *Plasmodium* infection. Peripheral blood (5 ml in EDTA) was collected from each subject at the time of diagnosis and additionally at after 7, 14, 21 and 28 days for cohort 1. Peripheral blood mononuclear cells (PBMC) were isolated using Ficoll-Paque density gradient system (Sigma). For all patients and control subjects of cohort 2, analysis of biochemical and cellular parameters was performed and demographic and epidemiological data were collected.

**Table 4. Anemia status of patients from cohorts 1 and 2.**

| Grades of Anemia [26] | Number of patients (%) | | | |
|---|---|---|---|---|
| | Cohort 1 | | Cohort 2 | |
| | *Pv* | *Pf* | UM | CM |
| No anemia (>11.9g) | 3 (27%) | 4 (44%) | 21 (36%) | 18 (36%) |
| Mild (11.9g/dL—10.9g/dL) | 8 (73%) | 5 (55%) | 21 (36%) | 12 (24%) |
| Moderate (10.8g/dL—8g/dL) | 0 | 0 | 17 (28%) | 19 (38%) |
| Severe (<8g/dL) | 0 | 0 | 0 | 1 (2%) |

Abbreviations: *P. vivax (Pv)*, *P. falciparum (Pf)*, uncomplicated *P. vivax* malaria (UM); complicated *P. vivax* malaria (CM).

## Determination of antibodies

Costar 3750 96-well ELISA plates were coated with PS (Sigma) at 20 μg/ml in 200-proof Molecular Biology ethanol or a lysate of freeze-thaw control human red blood cells (RBC) (Interstate Blood bank) at ($10^9$ RBCs/μl), calf thymus DNA (Sigma) at 10μg/ml and recombinant *P. vivax* MSP-1$_{19}$ (BEI resources, MRA-60) in PBS. Plates were incubated during 16 h at 4 ˚C (ethanol evaporates completely). Plates were washed 3 times with PBS 0.05% Tween-20 and then blocked for 1 h at 37 ˚C with PBS 1X 3% BSA buffer. Plasma was diluted at 1:100 in blocking buffer and incubated for 2 h at 37 ˚C. Plates were washed again 3 times and incubated with a polyclonal sheep anti-human IgG-HRP diluted 1:2000 (GE Healthcare) in PBS 1X 0.5% BSA for 1 h at 37 ˚C. Plates were washed 3 more times and developed using TMB substrate (BD Biosciences). The reaction was stopped using Stop buffer (Biolegend) and absorbance read at 450nm. The mean OD at 450nm from replicate wells was compared with reference serum from a Colombian *P. vivax* patient previously identified as high responder for anti-PS IgG antibodies to calculate relative units (RU). A healthy USA control was used as a negative control to assess background, in addition to the uninfected endemic Colombian controls. ELISA methods were done as previously described [15, 16].

## Flow cytometry

All flow cytometry was performed on a FACSCalibur (Becton Dickinson, Franklin Lakes, NJ) and analyzed with FlowJo (Tree Star, Ashland, OR). All Abs for FACS were purchased from BioLegend (San Diego, CA). PBMC were stained with anti-human: FITC anti-CD20 (2H7), PE anti-T-bet (4B10), FITC anti-CD11c (3.9), FITC anti-CD27 (O323), FITC anti-CD21 (Bu32), APC anti-CD21 (Bu32), APC anti-FcRL5 (509f6), APC anti-CD10 (HI10a), and PRCP anti-CD19 (HIB19). Intracellular T-bet staining was performed using the True-Nuclear Transcription Factor Buffer Set (Biolegend) and following manufacturer's instructions. Two to three technical replicates (independent labeling of PBMC and FACS analysis) for B-cell subpopulations were performed when the number of PBMC collected from each patient allowed for it (80 samples for cohort 1 and 30 samples for cohort 2). The same 8 uninfected non-endemic Colombian controls from Monteria were used for both cohorts. The average value of technical replicates for each sample was used for statistical analysis.

## Nadir calculation

The lowest hemoglobin level reading during the longitudinal time series for each patient was chosen as the nadir. For all correlations of cohort 1, two anemic time points were used.

## Statistical analysis

Data were analyzed using Prism (GraphPad Software). Student t-tests or One-Way Anova were used to identify statistical differences between groups of samples. A p-value of $<0.05$ was considered significant. Correlations were performed using non-parametric Spearman correlation analysis. Error bars represent the standard deviations (SD) of data from all of the patients used in each analysis.

## Results

### Levels of autoantibodies between uncomplicated *P. falciparum and P. vivax,* infections

In the first cohort, plasma samples from 20 patients with either *P. vivax* (n = 11) or *P. falciparum* (n = 9) uncomplicated infections were collected at the day of diagnosis (day 0), before patients received their first dose of treatment, and weekly during one month (days 7, 14, 21 and 28 (n = 99 unique samples). A large proportion of patients presented mild anemia (72.7% in *P. vivax* and 55.5% in *P. falciparum* patients) (Table 4). Both plasma and PBMC samples were also collected from uninfected healthy Colombians once (n = 8) (Table 1). First, we characterized the levels of relevant autoantibodies (PS, RBC and DNA) in *P. vivax* and *P. falciparum* patients across the different follow up samples at two time points where patients presented anemia. We observed a significant increase in the levels of all autoantibodies in both *P. falciparum* and *P. vivax* infections (Fig 1A–1C).

### Longitudinal analysis of autoimmune antibodies, atypical memory B cells and hemoglobin levels in *P. vivax* patients

We first analyzed the dynamics of autoantibodies and hemoglobin in the longitudinal samples from the first cohort, most of which suffered mild anemia (Table 4). The levels of hemoglobin varied over time for each patient, with most patients showing an initial decrease for 1–2 weeks until reaching the lowest hemoglobin concentration, or nadir, and a few showing a continuous increase in hemoglobin levels until recovery (n = 2 in *P. vivax* group) or an initial recovery followed by a later decrease (n = 2 in *P. falciparum* group). For this reason, the data were analyzed considering the nadir as the reference time, which is reached within one or two weeks difference between patients (Fig 2). Importantly, there was no association between the levels of hemoglobin and parasitemia (S2 Fig).

Analysis of the levels of autoimmune antibodies in all *P. vivax* samples (n = 52 unique samples) revealed that anti-RBC, anti-PS and anti-DNA IgG antibodies follow an inverse pattern compared with hemoglobin levels, showing their highest levels close to the time of hemoglobin nadir (Fig 2A, 2C and 2E). There was no significant difference between overall levels of autoantibodies between *P. vivax* and *P. falciparum* samples (Fig 1). Anti-MSP1 antibodies also declined at the last follow up (S3A Fig). The dynamics of all *P. falciparum* samples (n = 47 unique samples) followed similar trends where autoantibodies levels were inverse to hemoglobin and started declining after the nadir (day 7) which correlated with initial hematological recovery (Fig 2B, 2D and 2F). This initial dynamic analysis suggests a relationship for these autoantibodies and anemia.

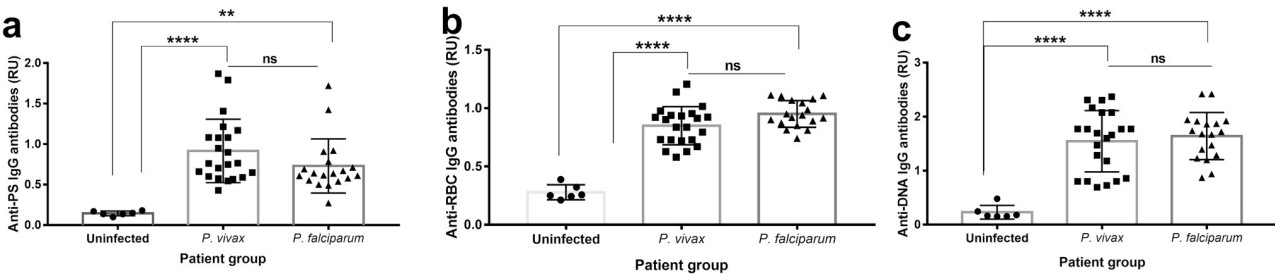

**Fig 1. Levels of autoantibodies are increased in *P. falciparum* and *P. vivax* malaria patients.** Bar graphs representing the levels of anti-PS (a), anti-RBC lysate (b) and anti-DNA (c) IgG antibodies at anemic time points between *P. vivax* and *P. falciparum* patients from cohort 1. Significance assessed by One-way Anova. $^*p \leq 0.05$, $^{**}p \leq 0.01$, $^{****}p \leq 0.0001$.

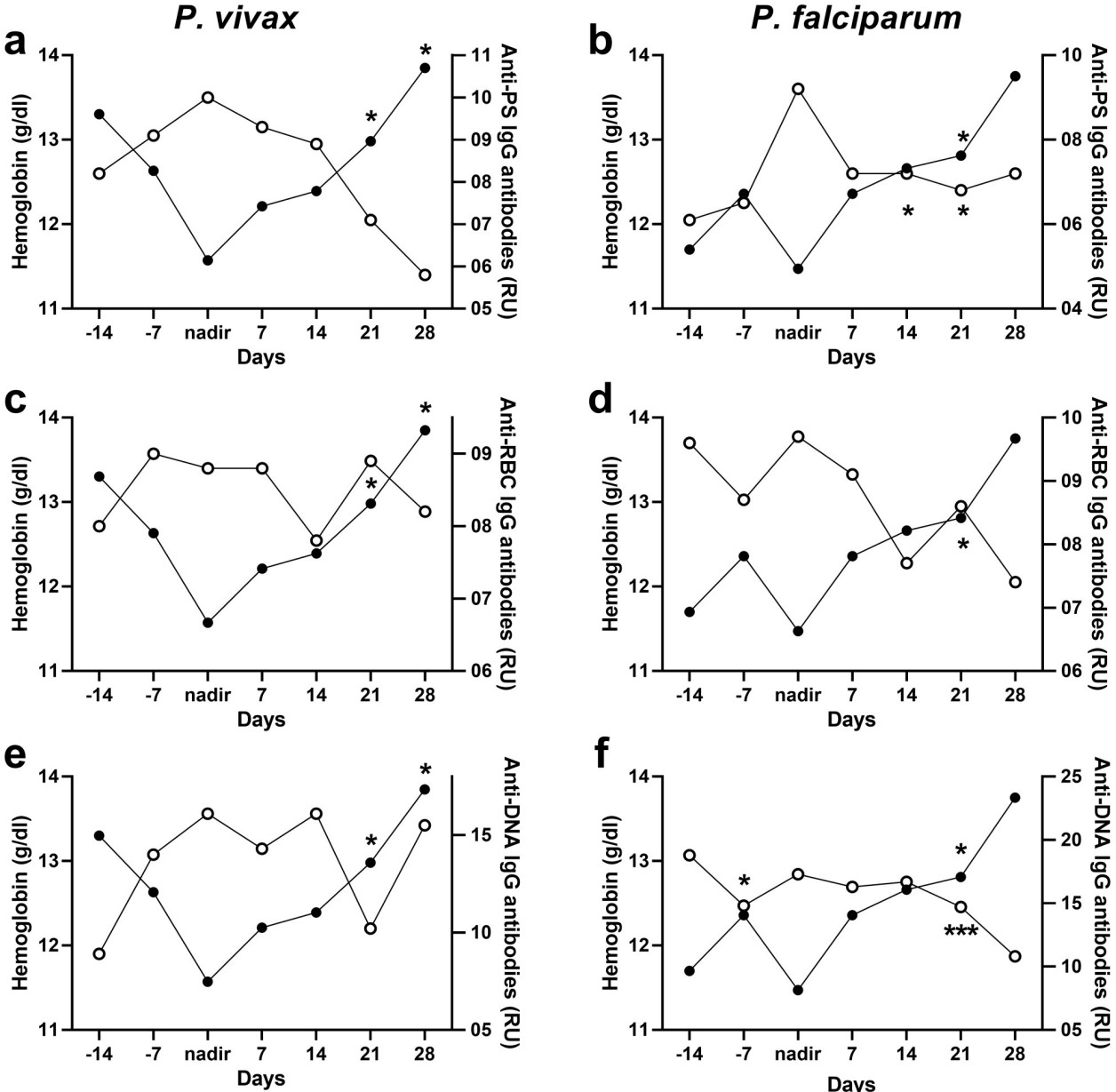

**Fig 2. Dynamics of autoantibodies and hemoglobin levels in Colombian *P. vivax* and *P. falciparum* patients.** Longitudinal analysis of hemoglobin (black circles) and autoantibody (white circles) dynamics of Colombian *P. vivax* (a,c,e) and *P. falciparum* patients (b, d, f) normalized to the nadir in each patient. Significant assessed by paired Student T-test for differences for each point with the nadir value are indicated $^*p \leq 0.05$, $^{***}p \leq 0.005$).

Correlation analysis of hemoglobin levels and different autoantibodies revealed an inverse relationship with anti-PS IgG antibodies (Fig 3A and 3B) and other autoantibodies (Fig 3C–3F) in both *P. vivax* and *P. falciparum*, suggesting a role in promoting anemia. As control, we observed that anti-*P. vivax* MSP1 IgG antibodies did not correlate with hemoglobin (S3C and S3D Fig), highlighting the specificity of the correlation with autoimmune antibodies.

We then analyzed the different B-cell populations in this first cohort of patients in peripheral blood mononuclear cells (PBMC) samples obtained at the same times as the plasma

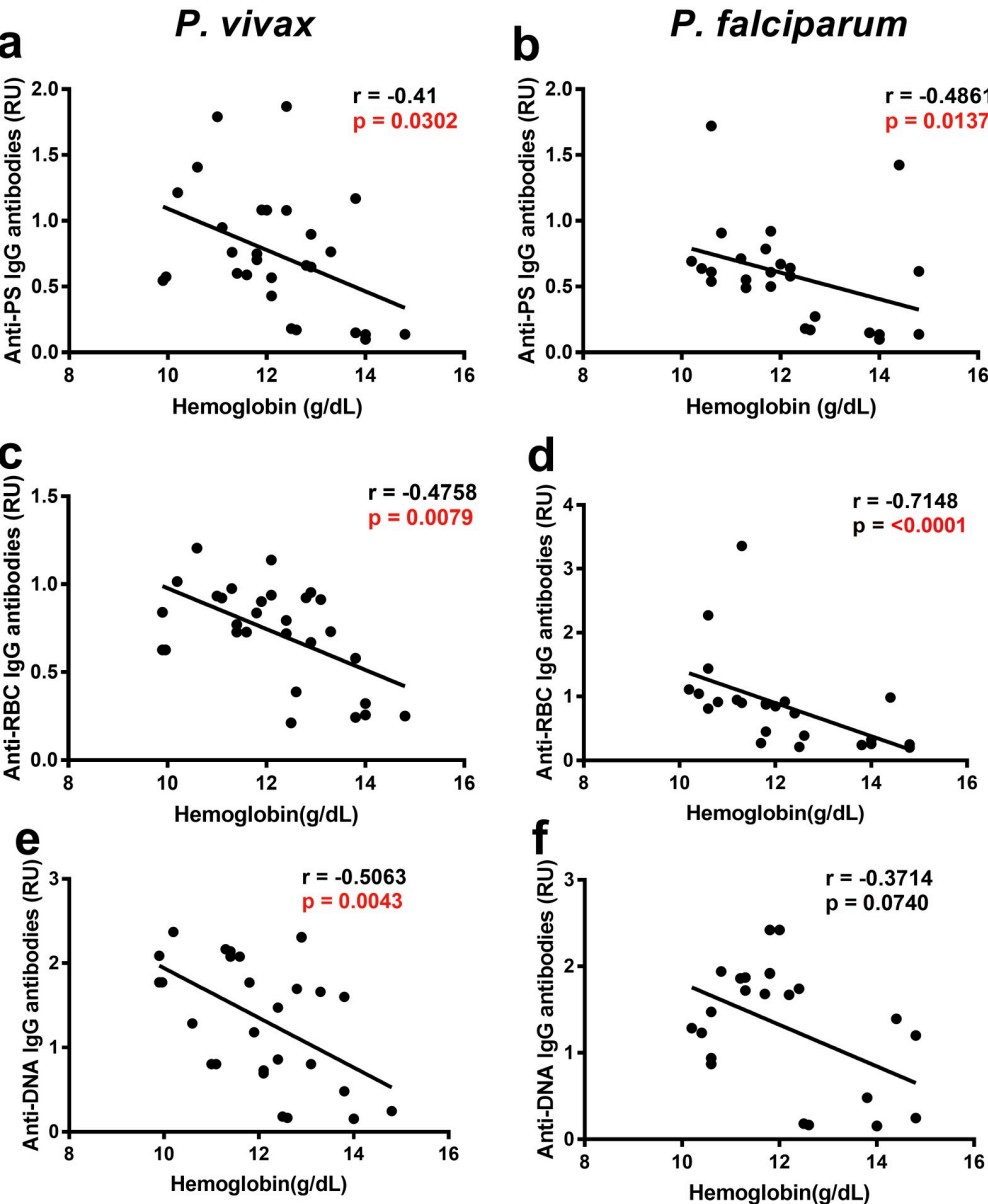

**Fig 3. Autoantibodies correlate with anemia in Colombian *P. vivax* and *P. falciparum* patients.** Correlation analysis of hemoglobin levels and autoantibodies of Colombian *P. vivax* (a, c, e) and *P. falciparum* patients (b, d, f) at anemic points. Significance was assessed by non-parametric Spearman correlation analysis.

(n = 48 for *P. vivax*, n = 32 for *P. falciparum* and n = 8 healthy controls). Following classical gating strategies for all relevant B-cell (CD19+) subpopulations from human PBMCs [16], we analyzed: (i) naïve B-cells (CD27−CD21+ CD10−), (ii) immature B-cells (CD10+), (iii) plasma cells (CD27+CD21−CD20−), (iv) classical MBCs (CD27+CD21+) and (v) atypical MBCs (FcRL5+T-bet+).

We first analyzed the levels of AtMBCs in this Colombian cohort finding significant increases in *P. vivax* and *P. falciparum* patients compared to healthy controls. The total levels of AtMBCs were similar in *P. falciparum* and *P. vivax* patients (Fig 4A). We then assessed the

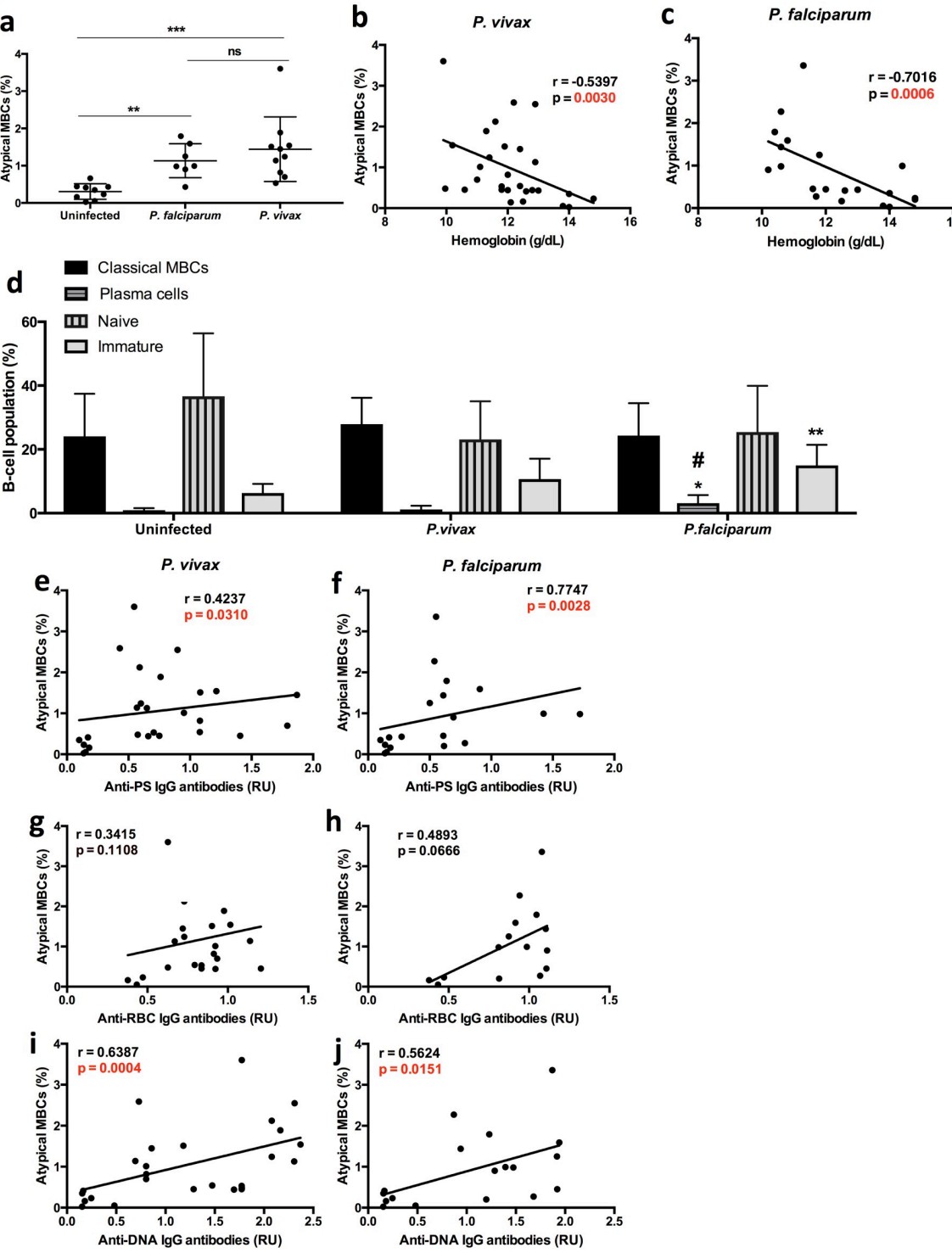

**Fig 4. Atypical memory B-cell correlate with hemoglobin in Colombian *P. vivax* and *P. falciparum* patients.** Graphs representing levels of atMBCs (a-c, e-j) or other B-cell populations (d) from PBMCs of Colombian uninfected controls, *P. vivax* and *P. falciparum* patients. Correlation analysis of atMBCs with either hemoglobin (b-c) or autoantibodies (e-j) of Colombian *P. vivax* (b, e, g, i) and *P. falciparum* patients (c, f, h, j) at anemic points. Significance assessed by One-way Anova (a,d) or by non-parametric Spearman correlation analysis (b-c, e-j). *p≤0.05, **p≤0.01, ***p≤0.005. #significant between *P. vivax* and *P. falciparum*.

relationship between atMBCs with anemia, finding a significant negative correlation with hemoglobin levels in both *P. vivax* and *P. falciparum* patients (Fig 4B and 4C). Immature B-cells were significantly expanded only in *P. falciparum* patients compared to controls (Fig 4D), while plasma cells were the only B-cell population significantly more expanded in *P. falciparum* compared to *P. vivax* patients and controls. We also observed that other antibody-secreting B-cell populations did not correlate with anemia in *P. vivax* or *P. falciparum* patients, except for a significant inverse correlation between immature B-cells and hemoglobin only in *P. falciparum* patients (S4 Fig).

We then assessed the relationship between the levels of atMBCs with plasma autoantibody levels. Correlation analysis showed a positive relationship between the levels of atMBCs for most autoantibodies for both *P. vivax* (Fig 4E, 4G and 4I) and *P. falciparum* (Fig 4F, 4H and 4J). We found no significant correlation between previous malaria episodes and any of the autoantibodies tested or atMBCs (S5 Fig).

## Analysis of clinical parameters and autoimmune antibodies in patients with uncomplicated and complicated *P. vivax* infections

In the second cohort, plasma samples from healthy community controls (n = 50), uncomplicated (n = 56) and complicated (n = 50) *P. vivax* infected patients were used for the determination of a battery of clinical and biochemical parameters. Among the criteria used for determination of complicated malaria, thrombocytopenia (platelets < 50,000/μl) was the most frequent (66% of complicated patients). In this group, most patients suffered from mild to moderate anemia (Table 4) where the level of hemoglobin ranged from 6.8 to 15.1 g/dL. The average (11.2 g/dL) was lower than in uncomplicated (11.5 g/dL) or control (12.6 g/dL) groups, but still significantly higher than the established criteria for severe anemia (hemoglobin < 8g/dL). As for the first cohort, plasma was also used to determine the levels of different autoimmune antibodies (anti-PS, anti-RBC and anti-DNA) and *P. vivax* anti-MSP1. For both uncomplicated and complicated *P. vivax* malaria patients, all autoantibodies were detected at higher levels than uninfected controls (Fig 5A–5C). Anti-PS IgG antibodies were significantly increased in complicated compared to uncomplicated malaria patients. Antibodies against MSP1 were not different between the uncomplicated and complicated *P. vivax* malaria groups, but were higher than the uninfected control group (S6 Fig).

Analysis of clinical parameters and autoantibodies identified different relations in the groups of uncomplicated and complicated *P. vivax* infections (Fig 5D and 5E). In the group of complicated *P. vivax* infections, we observed a relation between the three autoimmune antibodies and erythrocyte count, hemoglobin, or hematocrit levels, suggesting that autoimmune antibodies may be related to the loss of erythrocytes in this population. No correlation was observed between anti-MSP1 antibodies and hemoglobin levels. These relations were not observed in patients with uncomplicated infections or healthy controls.

## Autoimmune antibodies against PS and DNA correlate with hemoglobin levels in complicated *P. vivax* patients

Since autoimmune, and in particular anti-PS, antibodies have been proposed to contribute to malaria-induced anemia, we further analyzed the relation between autoimmune antibodies and hemoglobin levels in *P. vivax* malaria patients. Individual analysis revealed that in the group of uncomplicated patients, anti-RBC antibodies correlate inversely with hemoglobin levels (Fig 6A), but no correlation was found for anti-PS (Fig 6B), anti-DNA ($p = 0.65$) or anti-MSP1 ($p = 0.09$) antibodies. The level of parasitemia did not correlate with hemoglobin levels ($p = 0.85$), or with any of the antibodies.

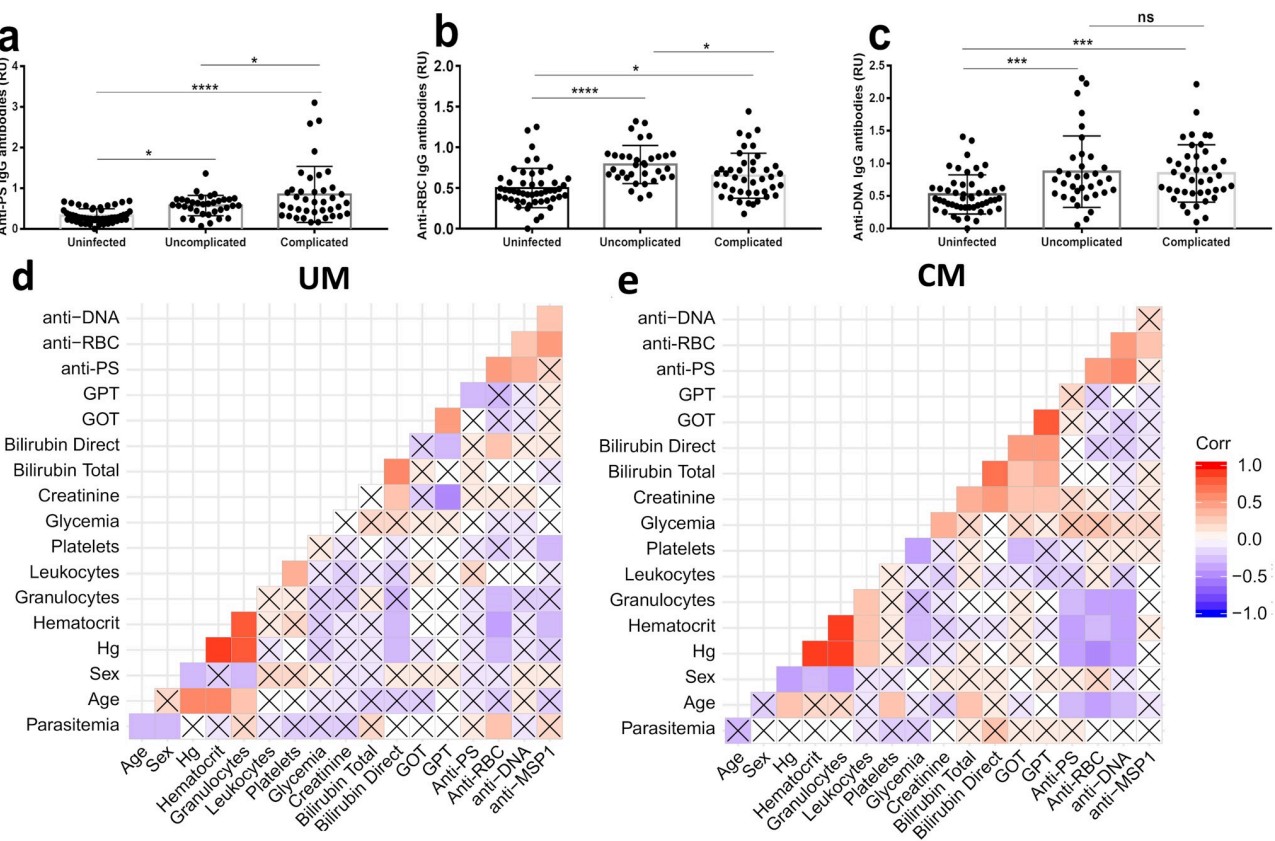

**Fig 5. Correlation analysis of autoantibodies with epidemiological and clinical parameters in *P. vivax* patients with uncomplicated or complicated malaria.** Bar graphs representing the levels of anti-PS (a), anti-RBC lysate (b) or anti-DNA (c) IgG antibodies in uninfected, uncomplicated or complicated *P. vivax* patients from cohort 2. A correlation matrix shows that anti-RBC, anti-PS and anti-DNA, but not anti-MSP1, correlate inversely with hemoglobin (Hg) in patients with complicated (CM) *P. vivax* (e), but not in uncomplicated (UM) (d). Spearman correlation coefficients are shown using the scale shown on the right, with positive correlations shown in red and negative correlations shown in blue. Boxes are marked with an "X" to show that the p-value for these pairwise correlations was >0.05. Significance assessed by One-way Anova (a-c) or non-parametric Spearman Correlation (d-e). $^*p \leq 0.05$, $^{**}p \leq 0.01$, $^{****}p \leq 0.0001$.

The autoimmune antibody response of patients with complicated *P. vivax* infections (Fig 6C–6F) showed that all three autoimmune antibodies tested: anti-RBC, anti-PS and anti-DNA antibodies, correlate inversely with hemoglobin levels, establishing a relation between the malaria-induced autoimmune response and hemoglobin levels in this group. Antibodies recognizing the parasite antigen MSP-1 presented no correlation with hemoglobin levels.

No significant correlations were found between any of the antibodies determined and the levels of parasitemia in these patients (*p* values are 0.76 for anti-PS; 0.66 for anti-RBC; 0.83 for anti-DNA; 0.74 for anti-MSP1).

## Atypical memory B-cells expand more and correlate with hemoglobin levels in complicated *P. vivax* patients

We then analyzed the relation of different B cell populations, including atMBCs and hemoglobin levels in *P. vivax* patients. First, we analyzed the levels of all relevant B-cell populations as described above (Fig 4), in uninfected healthy controls (n = 8), uncomplicated (n = 12) and complicated (n = 18) *P. vivax* patients (Fig 7). AtMBCs were significantly more expanded in

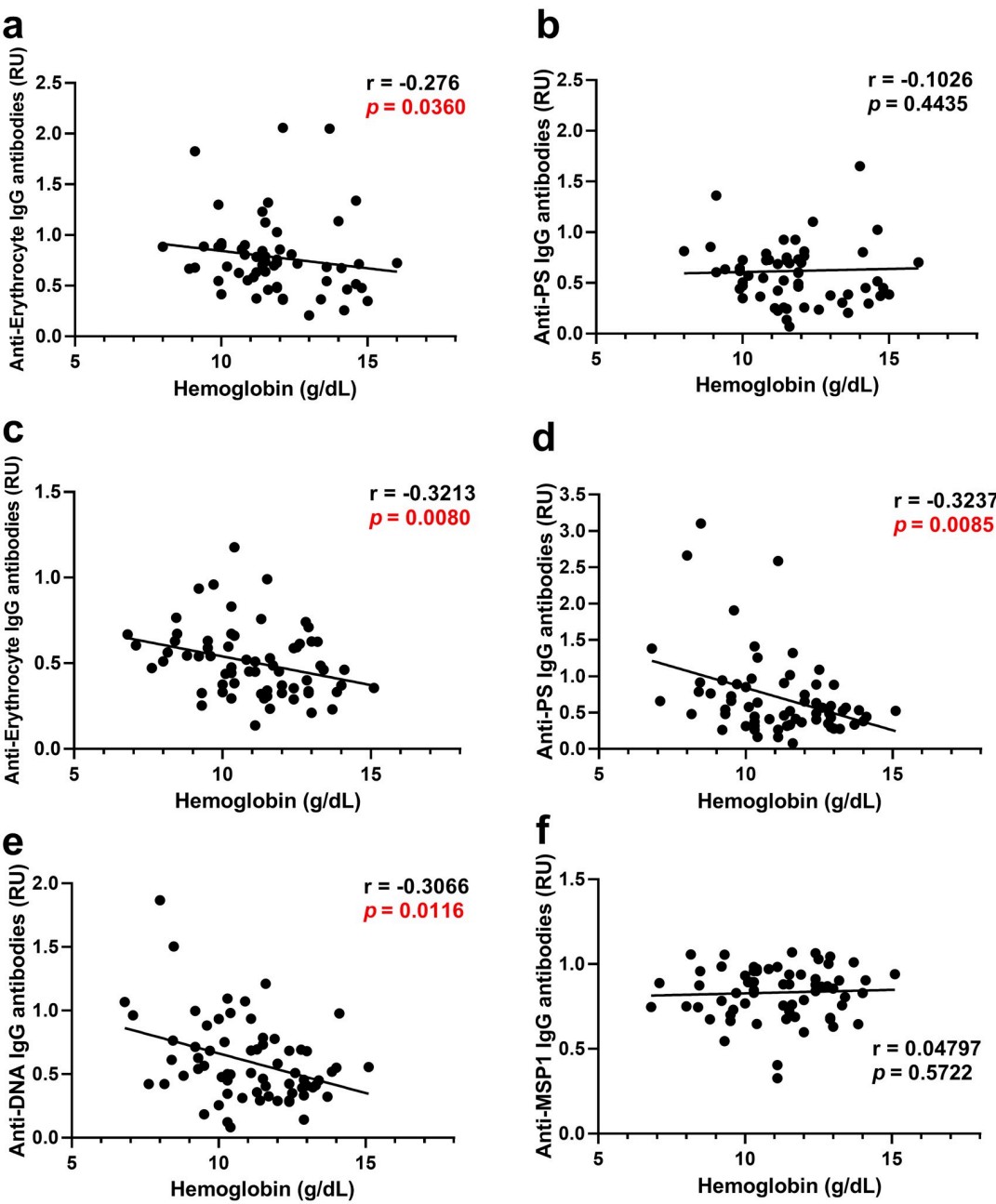

**Fig 6. Autoantibodies correlate with anemia development in complicated *P. vivax* patients.** Correlation analysis of hemoglobin levels and autoantibodies of Colombian *P. vivax* patients with uncomplicated (a-b) and complicated infections(c-f). Significance was assessed by non-parametric Spearman correlation analysis.

complicated compared to uncomplicated *P. vivax* patients (Fig 7A) and were the only B-cell population analyzed that was different between the *P. vivax* complicated and uncomplicated groups, suggesting that atMBCs may play a role in the severity of disease. Levels of immature B-cells were different when comparing uninfected controls and uncomplicated *P. vivax* patients (Fig 7G). Naïve B-cells were significantly decreased only in the *P. vivax* complicated group, when compared to controls (Fig 7I).

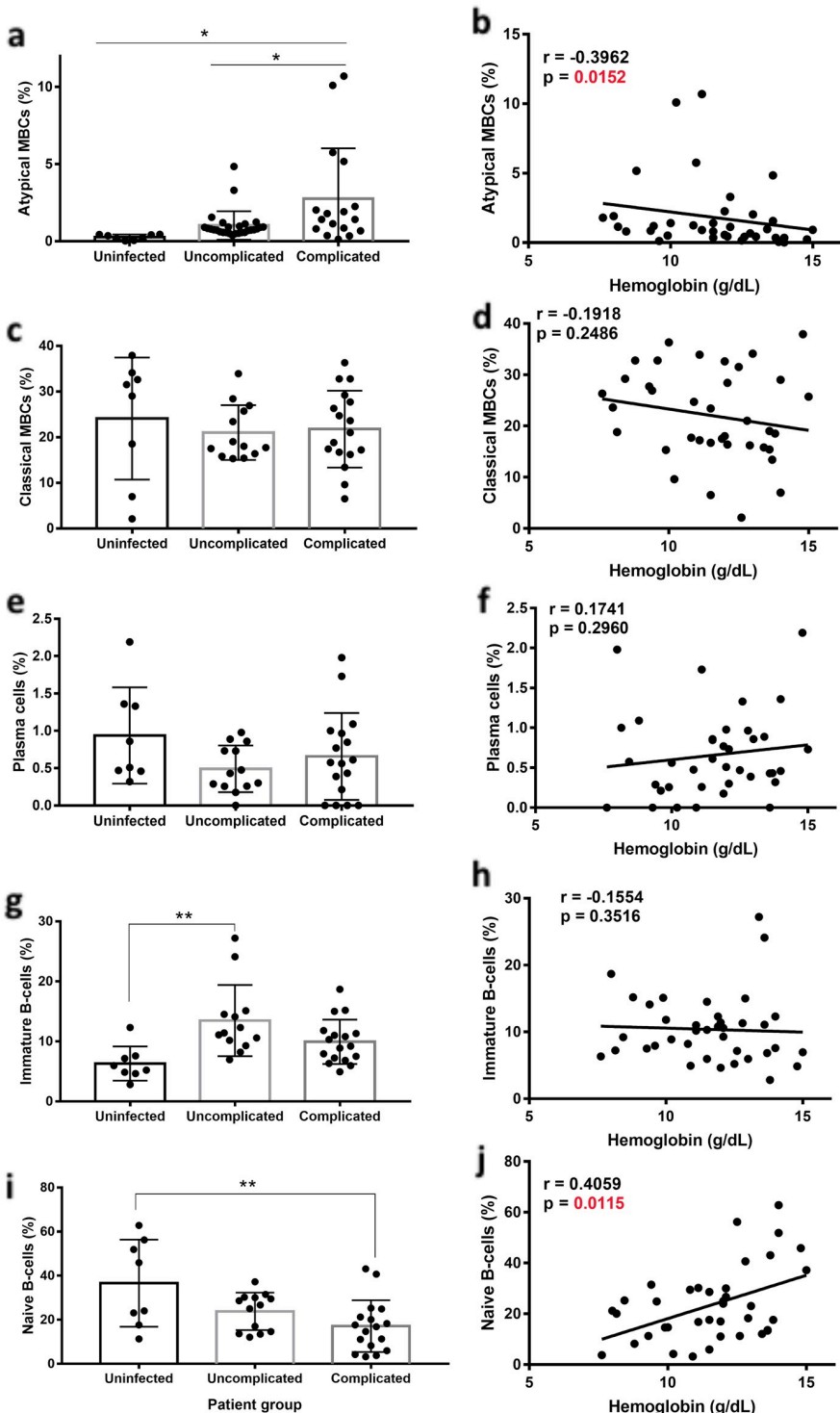

**Fig 7. atMBCs expand more robustly and correlate with hemoglobin in complicated *P. vivax* patients.** Percentage within CD19+ gate (a, c, e, g, i) and correlation with hemoglobin (b, d, f, h, j) of atMBCs (a, b), classical memory B-cells (c, d), plasma cells (e, f), immature B-cells (g, h) and naïve B-cells (i, j) from PBMCs of uninfected controls and *P. vivax* patients with uncomplicated or complicated infections. Significance assessed by One-way Anova (a, c, e, g, i) or non-parametric Spearman correlation analysis (b, d, f, h, j). *p≤0.05, **p≤0.01.

We then analyzed whether any B-cell populations were related to hemoglobin levels in *P. vivax* infections. Analysis of all B-cell subtypes analyzed showed that only atMBCs presented a significant inverse correlation with hemoglobin (Fig 7B), suggesting a role for these cells in malaria-induced anemia. There was a positive correlation between naïve B-cells and hemoglobin (Fig 7J), which is in agreement with the levels of these cells being significantly lower in complicated *P. vivax* patients (which also present lower hemoglobin levels) compared to healthy controls (Fig 7I). Other B cell populations did not show a relationship to hemoglobin levels (Fig 7D, 7F and 7H), underscoring the specificity of the atMBCs.

## Discussion

Complications during malaria caused by *P. vivax* is an increasingly reported phenomenon for which we lack understanding of its etiology [28, 29]. Anemia is one of most reported complications associated with *P. vivax* malaria, but little is understood about the mechanism leading to it [7, 30, 31]. In this study, we focused on characterizing the autoimmune B-cell response and its relation to malarial anemia in two different cohorts of malaria patients from Colombia, who suffered mostly from *P. vivax* malaria: one longitudinal in uncomplicated patients and one cross-sectional comparing uncomplicated and complicated patients. To our knowledge this is the first study to observe the presence and relationship of autoimmune antibodies, atMBCs and hemoglobin levels during *P. vivax* uncomplicated and complicated infections.

In the first cohort we observed a hemoglobin decrease in most patients for 1–2 weeks after treatment, which is consistent with previous studies in post-malarial anemia [32]. We found that the levels of three different autoantibodies, anti-PS, anti-RBC and anti-DNA, correlated negatively with levels of hemoglobin in both Colombian *P. vivax* and *P. falciparum* malaria patients. Anti-PS antibodies may promote malarial anemia by targeting for clearance newly born uninfected erythrocytes, called reticulocytes, which prematurely expose PS during malaria infection [10, 14, 33]. Since *P. vivax* preferentially infects reticulocytes[34], anti-PS antibodies could also target infected reticulocytes exposing PS, however no correlation was found between parasitemia and levels of any of the autoantibodies, suggesting that the role of autoantibodies in controlling parasite growth is not decisive in *P. vivax* malaria.

Binding of anti-PS antibodies to uninfected erythrocytes probably explains, at least in part, the correlation we observed between anti-RBC and hemoglobin, since PS is highly abundant in erythrocyte lysates, along with other reported protein auto antigens (spectrin and band 3) [12, 35]. Antibodies against DNA also correlate with anemia in Ugandan children who suffered *P. falciparum* malaria [15], but not in first-time infected European travelers [16]. The mechanism by which anti-DNA antibodies are related to anemia is not established but could be mediated by the recently reported ability of erythrocytes to bind cell-free DNA on their surface [10, 36]. Similarly as for other *P. falciparum* cohorts [15] [16], no correlation was observed between anti-parasite antibodies (*P. vivax* MSP1) and hemoglobin. However, a correlation between hemoglobin levels and different *P. vivax* antigens, including MSP-1, has been described in other cohorts with larger numbers of patients [37, 38]. The number of patients in our first cohort is relatively small, due to the difficulties in obtaining weekly samples from already recovered patients that do not require further medical attention. Similarly, the sample size of complicated *P. vivax* cases is limited by the relatively infrequent appearance of these cases. Despite these limitations, our study indicates a significant correlation between hemoglobin levels and autoimmune antibodies. If a weaker relation between hemoglobin and anti-MSP-1 antibodies exists, possibly was not observed due to the smaller sample size.

AtMBCs are a highly reported B-cell subset known to expand in *P. falciparum*-exposed individuals in endemic areas [18–21, 39] and in *P. vivax* patients [22, 40–42]. Importantly, we

have reported how *P. falciparum*-induced AtMBCs, characterized by double positivity of FcRL5 and T-bet, are able to secrete anti-PS antibodies *in vitro* and how they correlate with anemia in first-time infected *P. falciparum* patients [16]. This led us to explore whether AtMBCs also correlated with anemia and autoantibodies in Colombian *P. vivax* and *P. falciparum* malaria patients. Our results show an equally strong negative relationship between atMBCs levels and hemoglobin in both *P. vivax* and *P. falciparum* malaria patients. Lastly, plasma autoantibodies significantly correlated with levels of atMBCs, suggesting their association with anemia might be due to their ability to secrete these autoantibodies. No other antibody-secreting B-cell sub-population correlated with anemia development in these patients suggesting specificity for these cells and a role in promoting this syndrome. Altogether these data suggest a new role for atMBCs during anemia during *P. vivax* infections possibly through autoantibody secretion.

A surprising finding from this cohort is the similar levels of autoantibodies and atMBCs in *P. vivax* and *P. falciparum* malaria patients, while the loss of uninfected erythrocytes is known to be higher in *P. vivax* infections [9]. Our previous work in mice established a mechanism for the process of elimination of uninfected RBCs during malaria [14]. The elimination of uninfected RBCs depends directly on two factors: the exposure of PS on the surface of the RBC and the binding of anti-PS antibodies to it. A similar mechanism probably occurs in malaria patients, but the relative levels of PS exposure in RBCs during *P. falciparum* and *P. vivax* infections is not known. We acknowledge that the small sample number for this cohort and limiting the analysis to the hemoglobin nadir time point could be additional factors influencing these results. Nevertheless, our results show that atMBCs and autoantibodies are expanded and correlate with hemoglobin levels in both *P. vivax* and *P. falciparum* malaria patients.

In our second cohort, we compared the same parameters (anemia, atMBCs and autoantibodies) between uncomplicated and complicated *P. vivax* infections. In both groups of patients, some expected relations between hemoglobin and different leukocyte populations, age and sex [43, 44], were observed. The analysis of this second cohort revealed a strong negative correlation between all autoantibodies and hemoglobin specifically in complicated *P. vivax* infections. In patients with uncomplicated *P. vivax* infections, we did not observe a correlation of autoantibodies with hemoglobin levels. This difference in the results with cohort 1, where all *P. vivax* patients were uncomplicated but their autoantibody levels correlated inversely with hemoglobin, is most likely due to the fact that longitudinal data in cohort 1 allowed us to identify the hemoglobin nadir, which was used for the correlation analysis. The samples in cohort 2 are from a single time point, which probably does not coincide with the nadir in most patients. These suggests a temporal aspect of the role of autoantibodies in malarial anemia during *P. vivax* infections.

atMBCs were the only B-cell sub population that was significantly higher in complicated compared to uncomplicated *P. vivax* malaria patients, and was also highly correlated with hemoglobin levels in this cohort. Accordingly, anti-PS IgG antibodies were also significantly higher in complicated compared to uncomplicated *P. vivax* malaria patients. Since atMBCs are able to secrete anti-PS antibodies [16], their stronger expansion in complicated *P. vivax* infections could be directly linked to a pathological role. We observed a positive correlation between naïve B-cells and hemoglobin, which could be explained since both parameters had significantly decreased levels between complicated *P. vivax*-infected patients and uninfected individuals. Since naïve B-cells are not a source of antibodies [45], they probably do not play a role in autoimmune anemia. Altogether, these results further support a role for atMBCs and autoantibodies in mediating anemia and identify these atMBCs as possible indicators of complicated infections in *P. vivax* patients.

In summary, our results show the first evidence of atMBCs are correlated with autoantibodies and anemia during *P. vivax* malaria, particularly during complicated infections. Given the need for a better understanding of complicated *P. vivax* infections, atMBCs constitute a novel component to the complex etiology of this syndrome.

## Supporting information

**S1 Fig. Geographical location of sampling point at the Tierralta municipality in Cordoba department of Colombia.**
(TIF)

**S2 Fig. (Related to Figs 2 and 3). Parasitemia does not correlate with hemoglobin in *P. vivax* and *P. falciparum* Colombian patients.** Correlation analysis of initial parasitemia (day 0) with hemoglobin of Colombian *P. vivax* (a) and *P. falciparum* (b) patients. Significance assessed by non-parametric Spearman correlation analysis.
(TIF)

**S3 Fig. (Related to Figs 2 and 3). Dynamics of Anti-*P. vivax* MSP1 IgG antibodies in Colombian *P. vivax* patients.** (a-b) Longitudinal analysis of the dynamics of anti-*P. vivax* MSP1 IgG antibodies between day 0 and 28 post-treatment (a) and with hemoglobin (b). (c-d) Correlation analysis of anti-*P. vivax* MSP1 IgG antibodies for day 0 (c) and 28 (d) post-treatment with hemoglobin. Significance assessed by Unpaired Student T-test (a) and by non-parametric Spearman correlation analysis (c-d). *p≤0.05.
(TIF)

**S4 Fig. (Related to Fig 4). Levels of other B-cell populations in *P. vivax* and *P. falciparum* Colombian patients.** Correlation analysis of hemoglobin levels and classical memory B-cells (a-b), immature (c-d), naïve B-cells (e-f) and plasma cells (g-h) from PBMCs of *P. vivax* (a,c,e, g) and *P. falciparum* (b,d,f,h) patients at the two time points with lowest hemoglobin. Significance was assessed by non-parametric Spearman correlation analysis.
(TIF)

**S5 Fig. (Related to Figs 3 and 4). Correlation of previous malaria episodes with autoantibodies and atypical memory B-cells.** Correlation analysis of previous malaria episodes with anti-PS (a, b), anti-RBC lysate (c, d) or anti-DNA (e, f) IgG antibodies or atMBCs (g, h) at anemic time points between *P. vivax* (a,c,e,g) and *P. falciparum* (b,d,f,h) patients from cohort 1. Significance was assessed by non-parametric Spearman correlation analysis.
(TIF)

**S6 Fig. (Related to Fig 5). Anti-*P. vivax* MSP1 levels in uninfected and *P. vivax* patients with uncomplicated and complicated infections.** Bar graphs representing the levels of anti-*P. vivax* MSP1 antibody levels from plasma of uninfected controls and *P. vivax* patients with uncomplicated or complicated infection. Significance assessed by One-way Anova.
(TIF)

## Acknowledgments

We would like to thank the study individuals and their families for participating in the study and the study team for their dedication. Also, the hospitals San Jerónimo de Monteria and San José de Tierralta for their cooperation and dedication of their clinical personnel. Lastly, thank you to Dr. Anton Goetz for his helpful discussions.

## Author Contributions

**Conceptualization:** Juan Rivera-Correa, Ana Rodriguez.

**Data curation:** Juan Rivera-Correa, Maria Fernanda Yasnot-Acosta, Nubia Catalina Tovar, María Camila Velasco-Pareja, Alice Easton.

**Formal analysis:** Juan Rivera-Correa, Maria Fernanda Yasnot-Acosta, Nubia Catalina Tovar, Alice Easton, Ana Rodriguez.

**Funding acquisition:** Maria Fernanda Yasnot-Acosta, Ana Rodriguez.

**Investigation:** Juan Rivera-Correa, Maria Fernanda Yasnot-Acosta, Nubia Catalina Tovar, María Camila Velasco-Pareja, Alice Easton.

**Methodology:** Juan Rivera-Correa, Nubia Catalina Tovar.

**Project administration:** Maria Fernanda Yasnot-Acosta, María Camila Velasco-Pareja.

**Supervision:** Juan Rivera-Correa, Maria Fernanda Yasnot-Acosta, Ana Rodriguez.

**Validation:** Juan Rivera-Correa, Maria Fernanda Yasnot-Acosta.

**Visualization:** Juan Rivera-Correa.

**Writing – original draft:** Juan Rivera-Correa, Ana Rodriguez.

**Writing – review & editing:** Juan Rivera-Correa, Maria Fernanda Yasnot-Acosta, Ana Rodriguez.

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
