## [Decision Letter · Decision Letter 0]

11 May 2020

Dear Dr Rivera-Correa,

Thank you very much for submitting your manuscript "Atypical Memory B-cells and autoantibodies correlate with anemia during Plasmodium vivax complicated infections" for consideration at PLOS Neglected Tropical Diseases. As with all papers reviewed by the journal, your manuscript was reviewed by members of the editorial board and by several independent reviewers. The reviewers appreciated the attention to an important topic. Based on the reviews, we are likely to accept this manuscript for publication, providing that you modify the manuscript according to the review recommendations. 

All three reviewers have contributed tremendous feedback, with detailed analysis and valuable comments on how to improve the manuscript. As you address the comments, please pay particular attention to their suggested improvements, some which are numerous but hopefully not too onerous.

Sincerely,

Donelly Andrew van Schalkwyk, Ph.D.

Guest Editor

Hans-Peter Fuehrer

Deputy Editor

All three reviewers have contributed tremendous feedback, with detailed analysis and valuable comments on how to improve the manuscript. As you address the comments, please pay particular attention to their suggested improvements, some which are numerous but hopefully not too onerous.

Reviewer's Responses to Questions

**Key Review Criteria Required for Acceptance?**

**Methods**

-Are the objectives of the study clearly articulated with a clear testable hypothesis stated?

-Is the study design appropriate to address the stated objectives?

-Is the population clearly described and appropriate for the hypothesis being tested?

-Is the sample size sufficient to ensure adequate power to address the hypothesis being tested?

-Were correct statistical analysis used to support conclusions?

-Are there concerns about ethical or regulatory requirements being met?

Reviewer #1: (No Response)

Reviewer #2: The manuscript by Rivera-Correa et al. points to an association of anti-PS autoantibodies and atypical memory B cells with anemia in two different cohorts of P. vivax patients, particularly in complicated P. vivax ones. They have now extended their previous work by showing a strong inverse correlation between different IgG autoantibodies (anti-PS, anti-DNA, and anti-RBC) and atypical memory B cells (atMBCs) with hemoglobin in both P. vivax and P. falciparum patients over time. Moreover, they reported a stronger association between hemoglobin levels, atMBCs, and anemia in complicated P. vivax patients compared to uncomplicated ones. Despite anemia is an important clinical manifestation of P. vivax infection, little is known regarding the mechanism underlying this disease outcome, thus the work is an important contribution to the field. However, there are several points that must be addressed to make it clearer and more robust.

- The two main species that cause human malaria, P. falciparum and P. vivax, present biological particularities influencing the pathogenesis of severe infections, mainly anemia. The destruction of uninfected RBCs plays a crucial role in the etiology of vivax malaria, more than in P. falciparum infections. Thus, we expected an increase of anti-PS autoantibodies during vivax malaria when compared to levels in P. falciparum-infected patients. However, according to figure S2, the authors observed similar levels of autoantibodies (anti-Ps and anti-RBC) between P. vivax and P.falciparum Colombian patients. Do the authors have some explanation for such findings? Interestingly, levels of antibodies against RBC were lower than those verified for PS (please see fig. S2). In the same direction, levels of atMBCs were not different between patients infected with P. falciparum and P. vivax (Fig. S4). Those unexpected results in relation to the biological features of both parasite species were poorly discussed and deserve to be properly addressed. Moreover, I strongly recommend the authors to include figures S2 e S4 as part of the Results section, adjusting other figures to make the results more concise and direct (please see my comments below). 

- The pivotal novelty of the manuscript is the analysis of complicated malaria patients regarding their anemia status. However, it is not clear the criteria used to determine anemia and its classification according to the intensity as mild, moderate or severe. How many patients presenting complicated manifestation of vivax malaria had anemia (mild, moderate or severe)? Such information is essential to understand the relationship between anemia and autoantibodies and atMBCs. 

- The Introduction Section is too repetitive and, in my opinion, needs to be more focused, pointing the main concerns. I advise the authors to define the hypotheses to be tested in relation to the two cohorts evaluated in the study. This makes it very easy to understand the results and develop an appropriate discussion.

 - I strongly recommend authors to supply titles and statistical analyses for Tables. Currently, nothing is provided. Some are impossible to understand without such information.

- The small sample size of patients enrolled in the Cohort 1 is worth addressing. This limitation of the research should be addressed in the Discussion aiming to explain the obtained results.

Reviewer #3: The objectives of the study are well articulated however a hypothesis is not clearly stated. The study aims to test whether an association previously found in P. falciparum also exists in P. vivax, but using two different designs (longitudinal and cross-sectional) and additional types of patients. The population was clearly described and appropriate for the objetives.

The study was exploratory and the sample size was not large but it appeared sufficient to ensure adequate power to address some but not all objectives. A sample size or power calculation was not done for a primary analysis (which was not defined). Some correlations or dot plots had few points and that is probably the reason while statistical significance was not achieved in some of the secondary analyses.

The laboratory methods require more clarification: the ELISA for autoantibodies does not indicate the use of negative controls and the serum dilution is quite low (1/100), therefore there is concern about specificity of the response vs background response.

The statistical analysis used are quite simple but incompletely described. In methods it says that correlation analysis was Spearman but later they report Pearson in one occasion; Anova is not mentioned but later used, etc - revise consistency. 

The study received ethical approval from the Colombian IRB where patients were recruited.

**Results**

-Does the analysis presented match the analysis plan?

-Are the results clearly and completely presented?

-Are the figures (Tables, Images) of sufficient quality for clarity?

Reviewer #1: (No Response)

Reviewer #2: The authors need to remove any discussion, description of methods, references or conclusion from the results. They also need supply titles and, statistical analyses for Tables. Figures should be improved to highlight the main evidence. Please, consider joining some figure referring to related results.

Reviewer #3: The analysis matches to a large extent the objectives presented (there was no proper analysis plan, just some simple methods listed).

Results are in general clearly and completely presented but since a lot of correlations are done, it needs to be better explained how they are interpreted. It appears the weight is given to the p values (highlighting in red) but it is better to give weight to the correlation coefficient (and state what intervals are considered relevant), and the scatter plots, rather than to the P value in such Spearman analyses but this is not well explained. In some cases a significant p value might not be convincing when looking at the r or scatter plots, while in others not significant p values can still appear significant looking at the correlation plot, only that too few samples were included. In general, however, the correlation coefficients are reasonable for the conclusions.

Some supplementary figures need to be polished in terms of format, e.g. spaces (between P. and falciparum or vivax), italics, etc. Tables ok.

**Conclusions**

-Are the conclusions supported by the data presented?

-Are the limitations of analysis clearly described?

-Do the authors discuss how these data can be helpful to advance our understanding of the topic under study?

-Is public health relevance addressed?

Reviewer #1: (No Response)

Reviewer #2: Please, see the general comments above.

Reviewer #3: In general, the conclusions are supported by the data presented.

The imitations of analysis are not clearly described and the discussion is rather short, this should be expanded. There is also no sufficienc discussion on the public health relevance of the findings or in how these data can be helpful to advance our understanding of the topic under study.

**Editorial and Data Presentation Modifications?**

Reviewer #1: (No Response)

Reviewer #2: Minor comments:

- Title: Include “anti-phosphatidylserine antibodies” since only those specific antibodies have been studied. 

- Line 29: ‘are’ do not fit in this sentence.

- Line 35: add Colombia before South America.

- Line 70: this sentence needs to be supported by a reference.

- Line 72: the study conducted by Mourao et al. 2018 should also be cited here.

- Line 104: The sentence “Very little is known about the pathogenesis of complicated P. vivax infections” must be removed.

-Line 109-111: Please, delete the affirmation since it was very premature and not directly related to the paper.

- Line 123: Figure 1 is necessary? In my opinion, it could be changed to Fig. S2 and S6, showing the levels of antibodies against PS, RBC, and DNA in each cohort. As I mentioned above, those are unexpected results that deserve attention and discussion. 

- Line 126: Plasmodium vivax is a species, not a strain. Please, correct it.

- Line 133: In my opinion, it would be more informative if the authors mentioned here criteria used to determine anemia and its classification according to the intensity as mild, moderate or severe. Besides this, it is also important to explain how the authors determined parasitemia.

- Lines 127-132: Is there any overlap among patients from the two different cohorts?

-Table 1 is not adequate and must be improved to provide more information about cohorts and patients. How many patients were classified as presenting mild, moderate and severe anemia? According to the text, only one patient had levels of hemoglobulin above 7g/dL, what about the other two?

- Line 137: “(hemoglobin ≤ 7gr/dL)” – remove “r”

- Line 141: Please, mention here what were the excluded infections.

- Lines 141-143: Wouldn’t it be better to use individuals who have never been exposed to malaria? Did the authors consider the last species infecting everyone? In an endemic area, how can you be sure that those individuals did not have any infection along the six months period? Did the authors also perform the diagnosis to discard a subpatent infection (as they could be asymptomatic individuals)? 

- Line 150: table 1 - Please, change “(11)” to (n = 11) and (n = 9)

- Line 156: table 2 – Did you evaluate the number of previous malaria episodes? In my opinion, this variable is important to understand different aspects of vivax malaria including the outcome of mild or severe disease. In this table, the authors should also include other information that was evaluated and is also important to characterize patients such as GPT, GOT, bilirubin direct, bilirubin total, creatinine, glycemia, platelets, leukocytes, erythrocytes, hematocrit, Hg. Do the authors have data about reticulocytes count? If yes, please include it.

- Line 174: Which was the temperature used for blockage? Please, describe the blocking buffer. 

- Line 176: Please, mention the solution in which the secondary antibody was diluted.

- Line 178-180: Please, indicate at least one reference supporting this method to determine autoantibody levels. 

- Line 198: For figure 2, the authors did not perform any statistical analysis. As they compared the measurements of Hb and antibodies for the same patients, before and after nadir, the statistical test must be paired. Please review the analysis and complete this information.

- Line 201: replace “form” to “from”

- Line 207-210: Please, clarify the number of studied individuals: “P. vivax (n =11) or P. falciparum (n = 9)” means 20 patients, not 17 as pointed. “(n = 99 unique samples)” – is this the total number of samples collected in different moments? Please, confirm this.

- Line 216: Since the authors are talking about proportion, it is better to mention the percentage in parenthesis rather than numbers (8/11 and 5/9).

- Line 227: Change figure legend to Dynamics of autoantibodies and hemoglobulin levels in Colombian P. vivax and P. falciparum patients. Note that several patients are not anemic according to WHO criteria.

- Lines 235-236: This statement seems to be not clearly depicted in the graphs. Please, adjust the information accordingly. 

- Line 243: In my opinion, the relationship between two variables is generally considered strong when their r value is larger than 0.7. This is not the case for values such as: 0.41, 0.48, 0.47, 0.50, 0.37. Therefore, the word strong should be removed. 

- Line 251: What does “the two time points with lowest hemoglobin” mean? Please, clarify. 

- Line 257: Correct to subpopulations.

- Lines 271: Delete “in”. 

- Line 301: What does it mean “possible relation”? There is a relation or not regarding the results presented. 

- Lines 312-313: Change figure legend to Correlation analysis of autoantibodies with epidemiological and clinical parameters in P. vivax patients with uncomplicated or complicated malaria.

- Line 355-356: Why did the n change from this assay to the latter one?

- Line 408-411: I missed some discussion about the relationship between reticulocytes and P. vivax. How can anti-PS autoantibodies influence on P. vivax infection, as this species exclusively invades reticulocytes? 

- Lines 416-417: There is some evidence in the literature showing increased anti-PvMSP1-19 levels, negatively correlated with a decrease in hemoglobulin levels see Sepulveda et al., 2016). Other P. vivax antigens have also been correlating to anemia as PvMSP3 (see Mourao et al., 2012). Consider discussing the results accordingly.

- Line 420: remove South America since the main goal is not to evaluate geographical parameters but, outcomes related to P. vivax species that can be extrapolated to other patients living in vivax endemic areas. 

- Line 450: “…atMBCs constitute a novel component with diagnostic and

therapeutic potential for this syndrome.” This statement should be moderated.

- Figure 2: Add the word P. vivax above its corresponding graphs (a, c, e) and P. falciparum above b, d, and e. Moreover, correct the Y-axis of graph c (anti-RBC instead of anti-PS).

- Figure 3: The same observation described in figure 2.

- Figures 4 and 5 should be jointed as well as Figures 8 and 9.

Reviewer #3: Editorial suggestions as well as relatively minor modifications of existing data that would enhance clarity:

Abstract requires revising of the language (use of punctuation, selection of words, grammar, etc)

Atypical Memory B cells do not need capital letters at the early paragraphs.

In introduction reviese the use of words to avoid repetition of some terms.

Line 70: introduce a reference at the end of sentence.

There are key relevant papers missing on the presence of atypical memory B cells and P. vivax malaria that should be added, one in Latin America and the other in Papua New Guinea.

Review the use of abbreviations throughout to avoid duplicating definitions.

**Summary and General Comments**

Reviewer #1: Anemia is a common symptom seen in both P. falciparum and P. vivax infections. It has long been known that the level of anemia is out of proportion to the number of erythrocytes that are lysed from parasite infection. Besides suppressed erythropoiesis there is a large component of lysis of uninfected erythrocytes which contributes to this anemia. This group, and others, have made significant recent strides in determining the cellular mechanisms behind this phenomenon in both mouse models and P. falciparum. This manuscript extends those findings to P. vivax, where lysis of uninfected vs infected erythrocytes might be even more skewed towards uninfected.

This paper reports on two cohorts of P. vivax patients – one uncomplicated longitudinal cohort of both P. vivax and P. falciparum uncomplicated cases and one cross-sectional cohort comparing uncomplicated to complicated P. vivax infection. Although not overly innovative (the techniques are all identical to those used in previous studies in P. falciparum) the study is meaningful and impactful. The work had not previously been done in P. vivax and similarity to P. falciparum should not have been assumed. 

The work is very nicely done. It is presented well and analysed appropriately. Several concerns are listed below:

Figure 2 – the number of time points on the x-axis is confusing as presented. The Methods state that samples were taken and admission and then at d7,14,21, and 28 days after. That should result in a total of 5 times points. Why are there 7 time points on these graphs?

Figure 2 shows that P. vivax has more anti-DNA autoantibodies than P. falciparum. This was also shown in the Barber et al JID paper and deserves comment in the Discussion section. 

Figure 3 – the correlation of anti-DNA autoantibodies to anemia for P. falciparum is quite different from the results seen in the paper previously published by this group in eLife. Comments should be made.

Correlations between autoantibodies and anemia are seen in uncomplicated P. vivax in Cohort 1, yet they are NOT seen in the UCM cases in Cohort 2. Is that because of the natural history of P. vivax disease and the correlation would have developed later in the disease process? Or due to correlations only being seen in Cohort 1 when looking at the two lowest Hb points? This should be discussed. A reader that does not spend significant time in the manuscript might be confused with the apparent contradiction. The difference was also not mentioned in Lines 435 and 436.

The major contribution of this paper is extension of similar findings from P. falciparum to P. vivax. Given such, there should be at least a paragraph in the Discussion to compare and contrast the mechanisms in the two species. There might be hints of differences in anti-DNA autoantibodies. These should either be explained away or described in detail. There also are often numbers quoted that P. falciparum lyses 8 RBCs per every infected RBC, whereas P. vivax lyses 34:1. These numbers could either be explained as significantly different – or described as likely to be ‘statistically insignificantly different’.

Minor: 

Line 136 – hypoglycemia, not just ‘glycemia’

Table 2 – Consistency is needed with “.” vs “,” in numbers as well as the number of significant digits reported.

Line 207 – does not add up

Line 296 66% vs 64% on line 135

Figure 2 – I believe that the right-hand y-axis is mislabeled in graph c – shouldn’t this be anti-DNA?

Line 326 Is that meant to be “complicated” rather than “uncomplicated”?

Figure 9 – x-axes should be harmonized. The graph for Atypical B cells goes from 6 to 14 while the others go from 5 to 20

Reviewer #2: Although the manuscript represents an important contribution to the field of knowledge, some flaws should be considered by the authors during resubmission: (i) absence of a clear criterion for the characterization of anemia (mild, moderate and severe) in the study population difficult the interpretation of the role of anti-PS autoantibodies and atMBCs during malarial anemia; (ii) absence of an adequate comparison and discussion of the immunopathological processes of anemia during P. falciparum or P. vivax malaria, also difficult the interpretation of the results.

Reviewer #3: Novelty: the immunopathological mechanisms being investigated by the paper, which generate from seminal work by the senior author's lab (relationship betweein autoimmune antibodies particularly anti-PS, anemia, atypical B cells and malaria) are less innovative as similar studies have already been done in P. falciparum and P. vivax. The novelty now is related to the longitudinal nature of the analysis and the association with symptomatology and severity to P. vivax that is a more neglected human infection.

PLOS authors have the option to publish the peer review history of their article (what does this mean?). If published, this will include your full peer review and any attached files.

Reviewer #1: No

Reviewer #2: Yes: Erika Martins Braga

Reviewer #3: No
---

## [Editor Report · Decision Letter 1]

9 Jun 2020

Dear Dr Rivera-Correa and Dr Rodriguez,

We are pleased to inform you that your manuscript 'Atypical memory B-cells and autoantibodies correlate with anemia during Plasmodium vivax complicated infections' has been provisionally accepted for publication in PLOS Neglected Tropical Diseases.

Best regards,

Donelly Andrew van Schalkwyk, Ph.D.

Guest Editor

Hans-Peter Fuehrer

Deputy Editor

---

## [Editor Report · Acceptance letter]

14 Jul 2020

Dear Dr. Rivera-Correa,

We are delighted to inform you that your manuscript, "Atypical memory B-cells and autoantibodies correlate with anemia during Plasmodium vivax complicated infections," has been formally accepted for publication in PLOS Neglected Tropical Diseases.

Best regards,

Shaden Kamhawi

co-Editor-in-Chief

Paul Brindley

co-Editor-in-Chief
